# High diversity in Delta variant across countries revealed by genome-wide analysis of SARS-CoV-2 beyond the Spike protein

Rohit Suratekar[1,†] ⓘD, Pritha Ghosh[1,†] ⓘD, Michiel J M Niesen[2] ⓘD, Gregory Donadio[2] ⓘD, Praveen Anand[1] ⓘD, Venky Soundararajan[1,2,*] ⓘD & A J Venkatakrishnan[2,**] ⓘD

## Abstract

The highly contagious Delta variant of SARS-CoV-2 has become a prevalent strain globally and poses a public health challenge around the world. While there has been extensive focus on understanding the amino acid mutations in the Delta variant's Spike protein, the mutational landscape of the rest of the SARS-CoV-2 proteome (25 proteins) remains poorly understood. To this end, we performed a systematic analysis of mutations in all the SARS-CoV-2 proteins from nearly 2 million SARS-CoV-2 genomes from 176 countries/territories. Six highly prevalent missense mutations in the viral life cycle-associated Membrane (I82T), Nucleocapsid (R203M, D377Y), NS3 (S26L), and NS7a (V82A, T120I) proteins are almost exclusive to the Delta variant compared to other variants of concern (mean prevalence across genomes: Delta = 99.74%, Alpha = 0.06%, Beta = 0.09%, and Gamma = 0.22%). Furthermore, we find that the Delta variant harbors a more diverse repertoire of mutations across countries compared to the previously dominant Alpha variant. Overall, our study underscores the high diversity of the Delta variant between countries and identifies a list of amino acid mutations in the Delta variant's proteome for probing the mechanistic basis of pathogenic features such as high viral loads, high transmissibility, and reduced susceptibility against neutralization by vaccines.

**Keywords** coronavirus; Delta variant; mutations; proteome; SARS-CoV-2
**Subject Category** Microbiology, Virology & Host Pathogen Interaction
**Mol Syst Biol. (2022) 18: e10673**

## Introduction

The ongoing COVID-19 pandemic has infected over 210 million people and killed nearly 4.5 million people worldwide as of August 2021 (COVID-19 map—Johns Hopkins Coronavirus Resource Center, https://coronavirus.jhu.edu/map.html). Throughout the pandemic, the SARS-CoV-2 virus has acquired novel mutations, and the US government SARS-CoV-2 Interagency Group (SIG) has classified the mutant strains as variant of concern (VOC), variant of interest (VOI), and variant of high consequence (VOHC) (CDC, 2021). The variants of concern (Alpha: PANGO lineage B.1.1.7, Beta: B.1.351, Gamma: P.1, and Delta: B.1.617.2), as of August 2021, are more transmissible, cause more severe disease, and/or reduce neutralization by vaccines and monoclonal antibodies (CDC, 2021; Tracking SARS-CoV-2 variants, https://www.who.int/en/activities/tracking-SARS-CoV-2-variants/). The Delta variant (PANGO lineage B.1.617.2), first isolated from India in October 2020 (Tracking SARS-CoV-2 variants, https://www.who.int/en/activities/tracking-SARS-CoV-2-variants/), has emerged as the dominant global variant alongside the Alpha variant (PANGO lineage B.1.1.7), with genome sequences deposited from 104 and 150 countries, respectively, in the GISAID database (Shu & McCauley, 2017) and has worsened the public health emergency [WHO press conference on coronavirus disease (COVID-19)—July 30 2021; COVID-19 Virtual Press conference transcript—July 12 2021 (https://www.who.int/publications/m/item/covid-19-virtual-press-conference-transcript---12-july-2021)].

Recent studies are reporting nearly 1,000-fold higher viral loads in infections associated with the Delta variant (preprint: Li *et al*, 2021) and reduced neutralization of this variant by vaccines (Bernal *et al*, 2021; Liu *et al*, 2021a; Mallapaty, 2021; Wall *et al*, 2021; preprint: Tada *et al*). The NCBI database lists 26 proteins (structural, non-structural, and accessory proteins) in the SARS-CoV-2 proteome (SARS-Co-2 protein datasets—NCBI Datasets, https://www.ncbi.nlm.nih.gov/datasets/coronavirus/proteins/) totaling 9,757 amino acids. These include four structural proteins (Spike, Envelope, Membrane, and Nucleocapsid), 16 non-structural proteins (NSP1–NSP16), and six accessory proteins (NS3, NS6, NS7a, NS7b, NS8, and ORF10). As of August 2021, the CDC identifies 11 amino acid mutations in the Spike protein of the Delta variant (CDC, 2021), and the functional role of the SARS-CoV-2 Spike protein mutations has been well studied

---

1  nference Labs, Bengaluru, Karnataka, India
2  nference, Cambridge, MA, USA
   *Corresponding author. Tel: +1 857 207 2169; E-mail: venky@nference.net
   **Corresponding author. Tel: +1 650 919 3642; E-mail: aj@nference.net
   †These authors contributed equally to this work

(Duan *et al*, 2020; Huang *et al*, 2020; Shang *et al*, 2020). However, the mutational landscape of the rest of the Delta variant's proteome remains poorly understood. Concerted global genomic data sharing efforts through the GISAID database (Shu & McCauley, 2017) have led to the availability of nearly 2 million SARS-CoV-2 genomes from over 175 countries/territories, thereby providing a timely opportunity to analyze the mutational landscape of SARS-CoV-2 variants across all the 26 proteins.

Here, we perform a systematic analysis of amino acid mutations across the SARS-CoV-2 proteome (26 proteins) for the variants of concern and identify that the Delta variant harbors the highest mutational load in this proteome. Interestingly, the Delta variant's proteome is also highly diverse across different countries compared to the Alpha variant. Our observations suggest the need to account for country-specific mutational profiles for comprehensively understanding the biological attributes of the Delta variant such as increased viral loads and transmissibility, and reduced susceptibility against neutralization by vaccines.

## Results

### Delta variant has highly prevalent mutations in the viral life cycle-associated Membrane, Nucleocapsid, NS3, and NS7a proteins

Currently, only the Spike protein mutations are being used in literature to define the SARS-CoV-2 variants of concern and interest (CDC, 2021; Tracking SARS-CoV-2 variants, https://www.who.int/en/activities/tracking-SARS-CoV-2-variants/). However, the analysis of 1.99 million genome sequences of SARS-CoV-2 from 176 countries/territories in the GISAID database (Shu & McCauley, 2017) revealed mutations in 52.3% of the 9,757-amino-acid-long SARS-CoV-2 proteome. In all, there are 8,157 unique mutations in 5,107 amino acids spanning 24 of the 26 SARS-CoV-2 proteins (Fig EV1). The 1,055 unique amino acid mutations across 617 positions in the Spike protein contribute to only 6.3% of the mutated SARS-CoV-2 proteome (617 mutated positions of the total 9,757 amino acids in the SARS-CoV-2 proteome). This emphasizes the need to study the mutational profile across all the proteins of SARS-CoV-2.

Of the 1.99 million SARS-CoV-2 genomes analyzed here, there are 198,460 genomes corresponding to the Delta variant from 104 countries. We identified seven highly prevalent mutations in the following proteins of the Delta variant: Membrane (I82T: 99.9%), Nucleocapsid (R203 M: 99.9%, D377Y: 99.6%), NSP12 (P323L: 99.9%), NS3 (S26L: 99.9%), and NS7a (V82A: 99.4%, T120I: 99.7%). Strikingly, all these mutations except P323L in NSP12 are nearly exclusive to the Delta variant compared to other variants of concern (Alpha, Beta, and Gamma variants of SARS-CoV-2) (mean prevalence$_{Delta}$ = 99.74%, mean prevalence$_{otherVariantsofConcern}$ = 0.12%) (Fig EV2, Appendix Table S1). Within the Spike protein, there are four such mutations (T19R, L452R, T478K, and P681R) as well (mean prevalence$_{Delta}$ = 99.86%, mean prevalence$_{otherVariantsofConcern}$ = 0.04%). In total, there are 10 mutations across the proteome that are characteristic of the Delta variant, which can serve as candidates for probing the mechanistic basis of the Delta variant's pathogenic features.

The known functional implications of Delta variant mutations include antibody escape (Chi *et al*, 2020; Li *et al*, 2020b; Liu *et al*,

2021b; preprint: Venkatakrishnan *et al*, 2021), high viral load (Plante *et al*, 2021), increased transmissibility (Li *et al*, 2021; preprint: Cherian *et al*), and infectivity (Zhang *et al*, 2020; Table 1). We have assessed the evolutionary conservation of the 10 characteristic Delta variant mutations using Consurf (Ashkenazy *et al*, 2016) —graded on a scale of 1 (variable) to 9 (conserved) (Table 2). Protein sequence homologs were retrieved using HMMER (Eddy, 2011) against the UniRef90 database (Suzek *et al*, 2015), and the multiple sequence alignment was built using MAFFT (Katoh *et al*, 2002). We found the R203 M mutation in the Nucleocapsid protein to be highly conserved across 139 homologous protein sequences from coronaviruses. This position is indeed functionally important and is involved in the increased spread of the virus (Syed *et al*, 2021). It might also alter the binding of the human 14-3-3 protein to the proximal phosphorylated residues, leading to changes in the subcellular localization of the viral protein (Surjit *et al*, 2005; Del Veliz *et al*, 2021). Similarly, we also found that the I82T mutation in the Data ref: Membrane protein, 2020 is highly conserved across 92 homologous protein sequences from coronaviruses. This functionally important residue might lead to altered glucose binding and uptake, as predicted previously in literature (Shen *et al*, 2021). The functional impact of the remaining eight mutations could not be

**Table 1. Functional implications of mutations in SARS-CoV-2 Delta variant.**

| Mutation | Functional domain/region | Is solvent accessible? | Functional implications |
|---|---|---|---|
| Spike E156G | N-terminal domain | Yes | Antibody escape (Chi *et al*, 2020; preprint: Venkatakrishnan *et al*, 2021) |
| Spike ΔF157 | | | |
| Spike ΔR158 | | | |
| Spike L452R | Receptor-binding domain | Yes | Antibody escape (Li *et al*, 2020b; Liu *et al*, 2021b) |
| Spike T478K | | | |
| Spike D614G | – | Yes | Increases spike density and infectivity of virion (Zhang *et al*, 2020), and viral replication (Plante *et al*, 2021) |
| Spike P681R | – | Yes | Increased transmissibility (preprint: Cherian *et al*; Scudellari, 2021) |
| M I82T | Membrane-spanning helix (TM3)(Shen *et al*, 2021) | Yes | More biologically fit, with altered glucose uptake during viral replication (Shen *et al*, 2021) |
| NSP12 P323L | – | Yes | Increased transmissibility (preprint: Wang *et al*, 2020) |

Mutations in the SARS-CoV-2 Delta variant with known functional implications.

**Table 2. Computational characterization of highly prevalent SARS-CoV-2 mutations, exclusive to the Delta variant.**

| Mutation | Secondary structure | Domain/Site | ConSurf grade | No. of protein homologs | Overall predicted change in protein function |
|---|---|---|---|---|---|
| Spike T19R | Loop | N-terminal domain (Data ref: Spike glycoprotein, 2020) | [a] | 150 (coronaviruses) | Altered antibody interactions (Data ref: Cerutti et al, 2020) |
| Spike L452R | Strand | Receptor-binding domain (Data ref: Spike glycoprotein, 2020) | 1 | | Potentially increases binding to the ACE2 receptor |
| Spike T478K | Strand | | 1 | | |
| Spike P681R | Loop | Proximal to furin cleavage site (Data ref: Spike glycoprotein, 2020) | 1 | | Altered cleavage by host furin (Hoffmann et al, 2020) |
| Nucleocapsid R203 M | Loop | Proximal to phosphorylation site (SR-rich domain) (Tung & Limtung, 2020; preprint: Yaron et al, 2020) | 9 | 139 (coronaviruses) | Increased spread of the virus (Syed et al, 2021) and altered interaction with the human 14-3-3 protein (Del Veliz et al, 2021) leading to changes in subcellular localization (Surjit et al, 2005) |
| Nucleocapsid D377Y | Loop | – | 1 | | Functional impact of the mutation is unclear |
| Membrane I82T | Helix | Transmembrane domain (Data ref: Membrane protein, 2020) | 7 | 92 (coronaviruses) | Altered glucose binding and uptake |
| NS3 S26L | Helix | Proximal to viroporin transmembrane domain (Data ref: ORF3a protein, 2020) | [a] | 135 (coronaviruses) | Altered ion channel activity leading to change in NLRP3 inflammasome activation (key component of host antiviral response) (Chen et al, 2019) |
| NS7a V82A | Loop | – | [a] | 150 (coronaviruses) | Functional impact of the mutation is unclear |
| NS7a T120I | Loop | Proximal to polyubiquitination site (Li et al, 2020a) | 1 | | Altered IFN-I response (Xia et al, 2020) |

The evolutionary conservation of the residues was analyzed using Consurf (Ashkenazy et al, 2016), and graded on a scale of 1 (variable) to 9 (conserved) by the program. Protein sequence homologs were retrieved using one iteration of HMMER (Eddy, 2011) (E-value ≤ 0.0001) against the UniRef90 database (Suzek et al, 2015), and the multiple sequence alignment was built using MAFFT (Katoh et al, 2002).
[a]Unreliable conservation score due to calculations performed on less than six non-gapped homologous sequences.

assessed due to low conservation. Further experimental validation of these functional effects is warranted for a better understanding of their physiological impact.

**Delta variant is variable across countries and has country-specific core mutations**

While the Alpha variant spread widely during the pre-vaccination phase of the pandemic (Tracking SARS-CoV-2 variants, https://www.who.int/en/activities/tracking-SARS-CoV-2-variants/; Ledford et al, 2020), the Delta variant emerged as a global strain during the vaccination period. Given that the extent of vaccination coverage is highly variable across countries (Holder, 2021), the selection pressure against the Delta variant is also likely to vary. To understand mutational profiles of SARS-CoV-2 variants of concern across countries, we generate "mutational prevalence vectors" for each country of occurrence and calculate their pairwise cosine similarities (Fig 1A, Materials and Methods). The cosine similarity

distributions for the Alpha and Delta variants are significantly different (Jensen–Shannon divergence = 0.21, 95% confidence Interval: [0.17, 0.24], $P < 0.001$). The mean and standard deviation (SD) of pairwise cosine similarity values for the globally dominant Alpha and Delta variants ($mean_{Alpha} = 0.94$, S.D. $_{Alpha} = 0.05$; $mean_{Delta} = 0.86$, S.D. $_{Delta} = 0.1$) show a significantly higher diversity in the Delta variant as compared to Alpha (Cohen's d = 1.17, 95% confidence Interval: [1.02, 1.28], $P < 0.001$; Fig 1B, Appendix Fig S1).

To determine mutations that can contribute to country-specific differences in the Delta variant, we identified the highly prevalent mutations at the country level ("country-specific core mutations") (Fig 2A; Materials and Methods). As an example, here we compare the country-specific core mutations in the United States ($Delta_{UnitedStates}$) and in India ($Delta_{India}$). $Delta_{UnitedStates}$ has 29 country-specific core mutations compared with 19 country-specific core mutations in $Delta_{India}$ (Fig 2B). Of these, 16 mutations are common, spanning structural proteins (Spike, Nucleocapsid, and

**Figure 1. Schematic overview of the study.**

A Generation of country-specific mutation prevalence vectors and calculation of pairwise cosine similarity. The study dataset, updated as of July 31 2021, with nearly 2 million sequences were retrieved from GISAID. For a variant of concern, mutational prevalence vectors were calculated for each country of their occurrence. For example, the Delta variant has been reported in 104 countries worldwide and harbors 6,916 unique mutations. Thus, we generate 104 mutational prevalence vectors with $1 \times 6,916$ dimensions and calculate the pairwise cosine similarities for $^{104}C_2$ (5356) combinations.

B Comparison of probability distributions of pairwise cosine similarity values for the Alpha and Delta variants. The cosine similarity distributions for the Alpha and Delta variants are significantly different (Jensen–Shannon divergence = 0.21, 95% confidence Interval: [0.17, 0.24], $P < 0.001$). The mean and standard deviation (SD) of pairwise cosine similarity values for the globally dominant Alpha and Delta variants show significantly higher values in the Delta variant as compared to Alpha and thus a higher diversity (Cohen's d = 1.17, 95% confidence Interval: [1.02, 1.28], $P < 0.001$).

## A Generation of country-specific vectors and calculation of pairwise cosine similarity

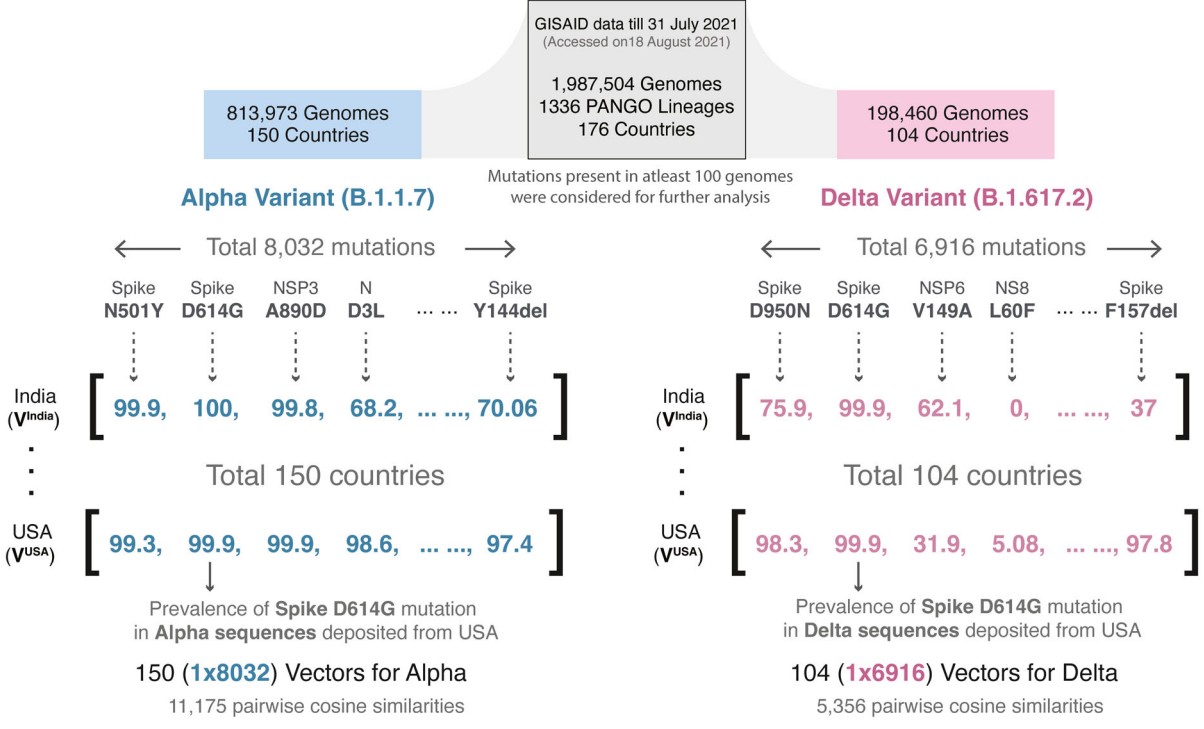

Cosine Similarity between India and USA = ( Dot Product of **V**$^{India}$ and **V**$^{USA}$ ) / (Product of magnitudes of **V**$^{India}$ and **V**$^{USA}$)

## B Comparison of pairwise cosine similarity distribution between Alpha and Delta

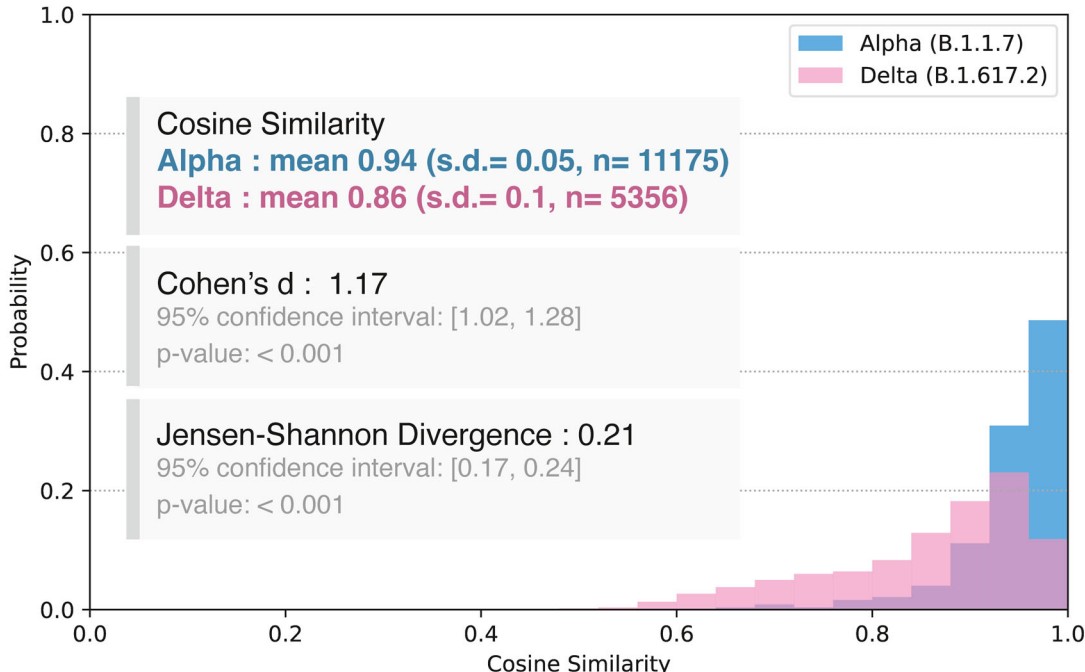

**Figure 1.**

## A  Method for calculating country-specific core mutations

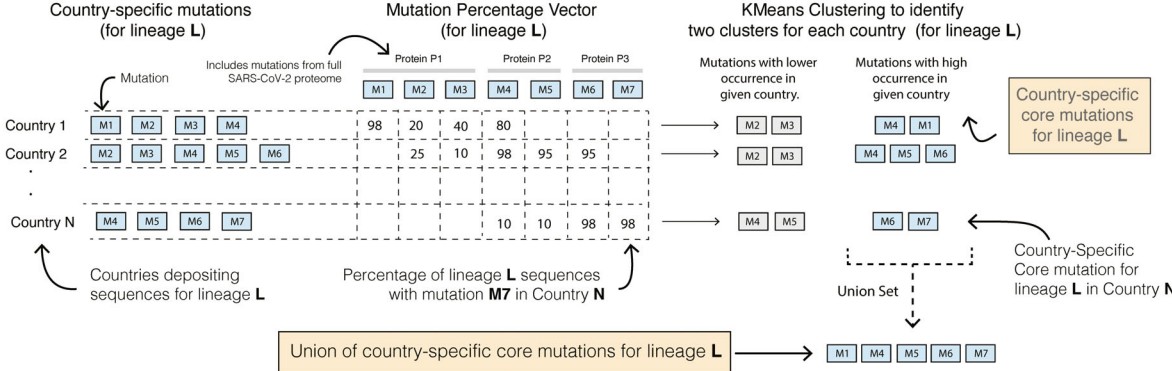

## B  Comparison of country-specific core Delta mutations between India and United States

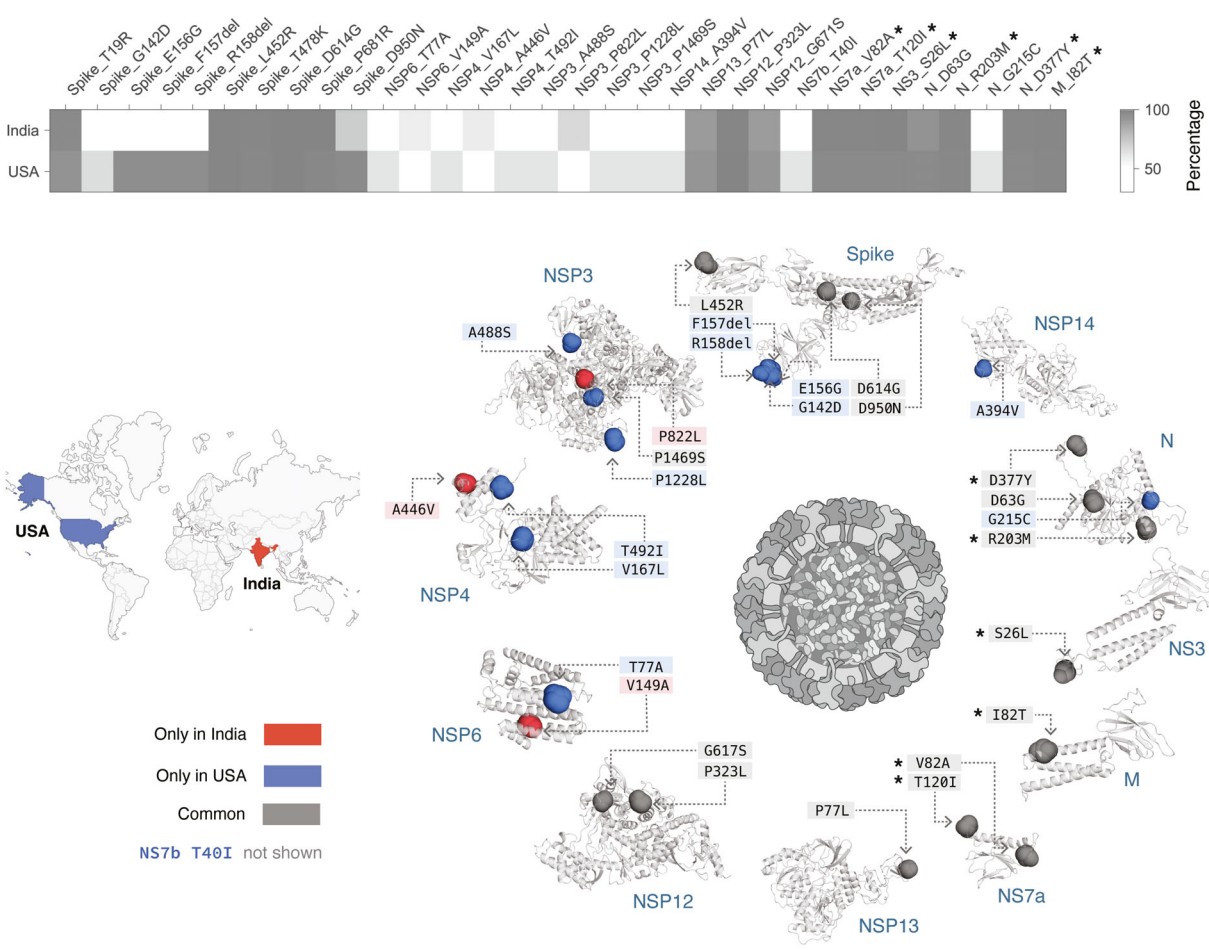

**Figure 2.  Identification of country-specific core mutations.**

A  Schematic overview of the method for defining country-specific core mutations for a lineage. See Materials and Methods for further details.

B  Comparison of prevalence of country-specific core mutations in the Delta variant in India and the United States. A total of 16 country-specific core mutations are common to both India and the United States, whereas 13 and 3 mutations are unique to the United States and India, respectively. The six mutations (in other SARS-CoV-2 proteins) marked with an asterisk are highly prevalent in all countries of occurrence of Delta variant (mean prevalence = 99.74%) but are nearly absent (mean prevalence = 0.12%) in the other variants of concern (Alpha, Beta, and Gamma variants of SARS-CoV-2). The mutations are highlighted on the structure of the Spike protein and the structural models of the other SARS-CoV-2 proteins (see *Methods*). Residues corresponding to Spike protein mutations T19R, T478K, and P681R are missing from the structure of the Spike protein and hence not shown here. The 43-amino-acid-long NS7b protein has no structure/model available and hence is not represented here.

Membrane), non-structural proteins (NSP3, NSP4, NSP6, NSP12, and NSP13), and accessory proteins (NS3 and NS7a).

There are three mutations in three proteins that are highly prevalent in Delta$_{India}$ but not in Delta$_{UnitedStates}$. In contrast, there are 13 mutations spanning six proteins that are highly prevalent in Delta$_{UnitedStates}$ but not Delta$_{India}$, including in the exoribonuclease NSP14, which is critical for the viral replication machinery (Ogando

et al, 2020) and can inhibit the host translational machinery (Hsu et al, 2021). We have assessed the evolutionary conservation of these mutations using Consurf, as described in the previous section (Appendix Table S2). We found the T492I mutation in the Nsp4C domain (possibly involved in protein–protein interactions; Data ref: Annotation rule, 2020) of the NSP4 protein is highly conserved across 139 homologous protein sequences from coronaviruses. This

**A   Hierarchical clustering of pairwise cosine similarities in Delta (B.1.617.2)**

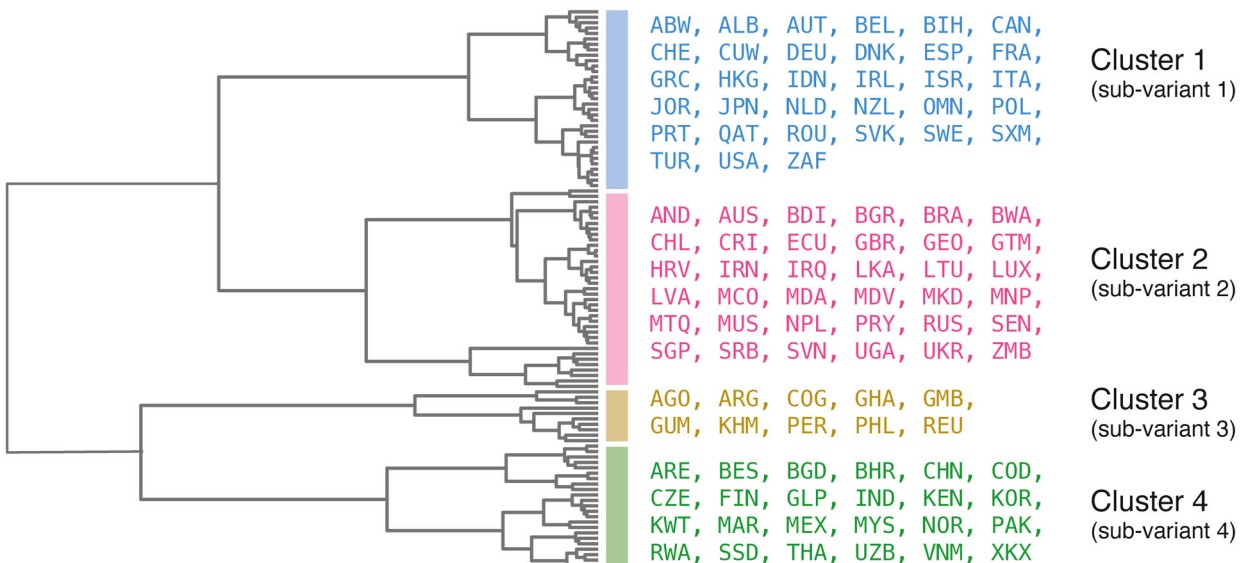

**B      Geographical distribution**

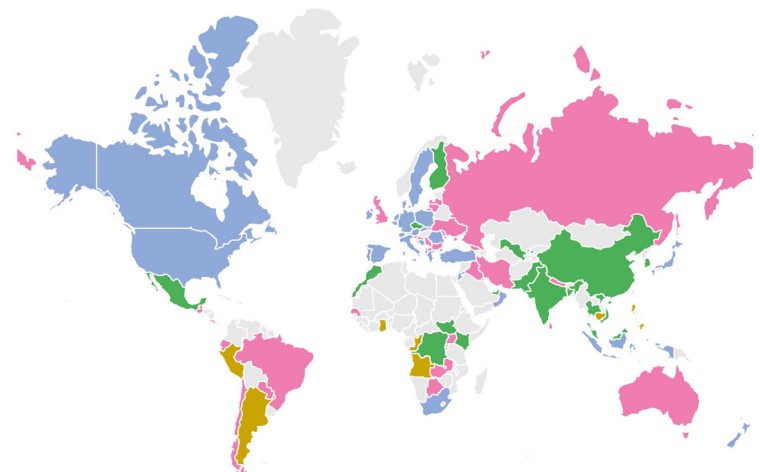

**Figure 3.   Comparison of the Delta sub-variants.**

A   Hierarchical clustering of pairwise cosine similarities across countries. We identified four clusters corresponding to four sub-variants of the Delta variant. The dendrogram shows the hierarchical relationship among the Delta sub-variants.

B   Geographical locations of the countries of localization of the sub-variants. The annotations on a map of the world show that the sub-variants are prevalent in geographically distant countries.

mutation can affect its interactions with protein-like ER homeostasis factors, N-linked glycosylation machinery, unfolded protein response-associated proteins, and antiviral innate immune signaling factors (Davies *et al*, 2020). The mutations in the functionally important positions in Delta$_{UnitedStates}$—Spike G142D, E156G, ΔF157, ΔR158 mutations—map to the antigenic supersite (Cerutti *et al*, 2021), possibly lead to immune evasion, and thus increase the virulence of this variant. The presence of country-specific differences in the Delta variants motivate the need to understand whether these genome-level differences manifest differences in the disease phenotypes and vaccine effectiveness.

## Discussion

COVID-19 is the first pandemic of the post-genomic era (van Dorp *et al*, 2021) that has been under intense genomic surveillance through concerted global viral sequencing efforts. This has led to the identification and tracking of emerging variants of concern, such as the highly transmissible Alpha variant and Delta variant. Through analysis of nearly 2 million genomes from 176 countries/ territories, we have identified that there are mutations beyond the Spike protein that are characteristic of the Delta variant and that the Delta variant is more variable across countries than other variants of concern.

Our study has identified 10 highly prevalent mutations characteristic of the Delta variant across five proteins, which can serve as therapeutic targets and as candidates for probing the mechanistic basis of the Delta variant's pathogenic features such as high viral loads, increased transmissibility, and reduced susceptibility against neutralization by vaccines. The country-specific differences in the Delta variant's mutational profile identified in this study can also be used to guide the design of vaccines/boosters that can comprehensively combat COVID-19. Our study also motivates that the diversity at the proteome level should be considered in designating the variants of concern and interest. This study shows that the sub-variants of the Delta variant (Fig 3A) are prevalent in geographically distant countries (Fig 3B), eliminating a causal relationship of geographical proximity with Delta variant diversity. However, future studies are warranted to comprehensively examine the combinations of factors such as vaccination rates, geographical proximity, and airline connectivity (Fig EV3) to dissect the difference in the epidemiology of Delta variants across countries.

This study has a few limitations. Since this study is based on publicly available data from the GISAID database, it may carry biases associated with sequencing disparities across countries and reporting delays. Although there is extensive genomic surveillance, there is a lack of clinical annotation of the genomes, limiting our ability to assess the clinical impact of the country-specific differences in the variants. The GISAID database does not record mutations in the recently discovered ORFs in the SARS-CoV-2 genome such as ORF10, ORF9b, and ORF9c. The assignment of the mutations in these ORFs may reveal further differences between SARS-CoV-2 variants.

Although mass vaccination efforts are underway around the world, there are huge differences in the population immunity of countries due to the differences in the vaccines approved regionally and the extent of vaccination coverage in populations. These differences contribute to the risk of emergence of new SARS-CoV-2 variants, which could pose challenges to existing therapies and vaccination (Weber *et al*, 2021). Continued genome surveillance is imperative for developing comprehensive global and country-specific preventive and therapeutic measures to end the ongoing pandemic.

## Materials and Methods

### SARS-CoV-2 genome sequences

We retrieved 1,987,504 SARS-CoV-2 high-coverage complete-genome sequences from human hosts in 176 countries/territories spanning 1,336 PANGO lineages on August 18 2021 from GISAID (Shu & McCauley, 2017) for December 2019 to July 2021, of which 816 sequences do not harbor any mutations. We removed sequences from other hosts and those with incomplete dates (YYYY-MM or YYYY) from further analyses. A total of 1,986,688 sequences harbor a total of 89,875 unique amino acid mutations. However, to account for errors arising from sequencing, we only consider 8157 unique mutations in 24 proteins that are present in 100 or more sequences for all our further analyses. We did not identify any mutations in NSP11 (for which no mutations are present in 100 or more sequences) and ORF10 (for which no information on mutations are available in GISAID data), and hence are not considered in further analyses.

Although 99.15% of all SARS-CoV-2 genome sequences possess one or more mutations in the Spike protein, 98.91% and 95.2% of sequences also bear mutations in the crucial NSP12 (RNA-dependent RNA polymerase, RdRp) and Nucleocapsid proteins, respectively.

We retrieved the list of proteins in the SARS-CoV-2 proteome from NCBI (SARS-CoV-2 protein datasets—NCBI Datasets, https:// www.ncbi.nlm.nih.gov/datasets/coronavirus/proteins/) on August 2 2021. The structure of the Spike protein was retrieved from PDB (code: 6VSB) and that of the structural models of the other SARS-CoV-2 proteins from https://zhanglab.ccmb.med.umich.edu/ COVID-19/ (on June 11 2021).

### Cosine similarity across countries

To calculate the cosine similarity of a lineage *L* among countries, we generated a prevalence vector of constituent mutations for each country of occurrence of the lineage *L*. For a pair of countries, the cosine similarity of the lineage *L* was calculated for their mutation vectors (A, B) (Equation 1, Fig 1A).

$$Cosine\ similarity\ (A, B) = \frac{A \cdot B}{|A| \times |B|} \qquad (1)$$

The mean and standard deviation (SD) of pairwise cosine similarity values for variants of concern (mean$_{Alpha}$ = 0.94, SD$_{Alpha}$ = 0.05; mean$_{Beta}$ = 0.89, SD$_{Beta}$ = 0.06; mean$_{Gamma}$ = 0.95, SD$_{Gamma}$ = 0.03; and mean$_{Delta}$ = 0.86, SD$_{Delta}$ = 0.1) show a higher diversity of the Delta variant across countries. To check the effect size, Cohen's d was calculated (Equation 2).

$$Cohen's \; d = \frac{M_2 - M_1}{\sqrt{\left(\frac{\left((n_1-1)\times SD_1^2\right) + \left((n_2-1)\times SD_2^2\right)}{(n_1+n_2-2)}\right)}} \qquad (2)$$

where $M$: mean, $n$: sample size, and SD: standard deviation.

Probability distributions of pairwise cosine similarities were calculated by binning frequencies (bins = 25), and their Jensen–Shannon divergence (with base 2) was calculated using the *jensen-shannon* function available in SciPy [v1.7.0] (Virtanen *et al*, 2020). $P$ was calculated using bootstrapping with 1,000 iterations.

To identify countries with similar mutational profiles, we clustered the pairwise cosine similarity matrix with Ward's variance minimization algorithm (Ward & Hook, 1963) available in SciPy [v1.7.0] (Fig 3A).

### Bootstrapping of cosine similarities

For each country, we resampled (with replacement) all the sequences deposited in the GISAID database and generated a cosine similarity distribution for Alpha and Delta variants (Fig EV4). For calculating 95% confidence interval, we calculated Jensen–Shannon divergence (JSD) and Cohen's d for each bootstrap iteration. To get a null distribution for JSD and Cohen's d, we calculated these metrics from the Alpha and Delta cosine similarity distribution generated in each bootstrap iteration ($n = 1,000$). The $P$-values were calculated based on the distribution of all bootstrapped values and original JSD/Cohen's d values.

### Cosine similarity for airline connectivity

Air traffic data were accessed on June 13 2021 from The OpenSky Network 2020 (Olive *et al*, 2021; Strohmeier *et al*, 2021). Only international flights were considered in this analysis. A matrix of the number of international flights across all countries of the world was generated for the period of February 2021 to June 2021. For country $A$, a vector of the number of outgoing flights to all the other countries normalized with respect to the total number of outgoing flights from country $A$ was generated. Similarly, for country $B$, a vector of the number of incoming flights from all the other countries normalized with respect to the total number of incoming flights to country $B$ was generated. Cosine similarity for airline connectivity for this pair of countries was calculated as in Equation 1.

### Country-specific core mutations

Genome sequences of Alpha, Beta, Gamma, and Delta variants in GISAID data are available from 150, 95, 61, and 104 countries, respectively. For country $C$, we calculated the prevalence of a mutation $M$ as in Equation 3.

Prevalence of $M(L|C)$

$$= \frac{\text{Number of sequences of lineage } L \text{ in country } C \text{ that harbor a mutation } M}{\text{Total number of deposited sequences of lineage } L \text{ in country } C}$$

$$* 100 \qquad (3)$$

The prevalence of all mutations identified in lineage $L$ in country $C$ was calculated and further clustered using K-means clustering algorithm (Lloyd, 1982) (in scikit-learn; Pedregosa *et al*, 2011) for unbiased identification of the highly prevalent set (core) of mutations for lineage $L$ in country $C$. Based on K-means clustering sensitivity analysis, we partitioned the observations into two clusters for K-means clustering with initial cluster centroids at 0% and 100% (Appendix Fig S2). All mutations with labels corresponding to the higher centroid are called the core mutations of lineage $L$ in country $C$ ("country-specific core mutations"). A union set of country-specific core mutations from all countries in which lineage $L$ is present were also determined. We observed that the Delta variant's union set of country-specific core mutations are distinct and higher from those in the other variants of concern (Fig EV5, Appendix Table S3).

The characteristic Spike protein mutations defined by the CDC (CDC, 2021) (as of August 2 2021) overlap with those identified in our analysis (Appendix Fig S3), thus validating our method of identifying mutations in the SARS-CoV-2 proteome.

## Data availability

This study includes no data deposited in external repositories.

**Expanded View** for this article is available online.

## Acknowledgements

The authors thank Murali Aravamudan, Arjun Puranik, Sutirtha Chakraborty, Gajinder Pal Singh, and Shahir Asfahan for feedback on this manuscript.

## Author contributions

**A J Venkatakrishnan:** Conceptualization; Supervision; Writing—review and editing. **Rohit Suratekar:** Data curation; Software; Formal analysis; Visualization; Methodology; Writing—original draft. **Pritha Ghosh:** Data curation; Formal analysis; Investigation; Methodology; Writing—original draft. **Michiel J M Niesen:** Data curation; Validation; Writing—original draft. **Gregory Donadio:** Resources; Software. **Praveen Anand:** Validation; Methodology; Writing—original draft. **Venky Soundararajan:** Conceptualization; Project administration.

In addition to the CRediT author contributions listed above, the contributions in detail are:
VS and AJV conceived the study. PG, RS, MJMN, and AJV designed the study, reviewed the findings, and wrote the manuscript. RS, PG, MJMN, GD, PA, AJV, and VS contributed to methods, data, analysis, or software. All authors revised the manuscript.

## Disclosure and competing interests statement

RS, PG, MJMN, GD, PA, VS, and AJV are employees of nference and have financial interests in the company and in the successful application of this research. nference collaborates with bio-pharmaceutical companies on data science initiatives unrelated to this study. These collaborations had no role in study design, data collection and analysis, decision to publish, or preparation of the manuscript.

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
