## [Review Process File · Molecular Systems Biology]

High diversity in Delta variant across countries revealed by genome-wide analysis of SARS-CoV-2 beyond the Spike protein

Rohit Suratekar, Pritha Ghosh, Michiel Nielsen, Gregory Donadio, Praveen Anand, Venky Soundararajan, A. J. Venkatakrishnan

DOI: 10.15252/msb.202110673

Corresponding author(s): A J Venkatakrishnan (aj@nference.net) , Venky Soundararajan (venky@nference.net)

Review Timeline:

Submission Date:	29th Sep 21
Editorial Decision:	29th Oct 21
Revision Received:	8th Dec 21
Editorial Decision:	16th Dec 21
Revision Received:	31st Dec 21
Accepted:	7th Jan 22

Editor: Jingyi Hou

Transaction Report:

29th Oct 2021

Manuscript Number: MSB-2021-10673

Title: High diversity in Delta variant across countries revealed via genome-wide analysis of SARS-CoV-2

Author: A J Venkatakrisnan

Rohit Suratekar

Pritha Ghosh

Michiel Nielsen

Gregory Donadio

Praveen Anand

Venky Soundararajan

Thank you for submitting your work to Molecular Systems Biology. We have now heard back from the three reviewers who agreed to evaluate your manuscript. As you will see from the reports below, the reviewers acknowledge the potential interest of the study. They raise, however, a series of concerns, which we would ask you to address in a revision of the current manuscript.

I think that the reviewers' recommendations are rather clear and there is no need to reiterate their comments. The issues raised by the Reviewers #1 and #3 need to be satisfactorily addressed. As you may already know, our editorial policy allows in principle a single round of major revision, and it is therefore essential to provide responses to the reviewers' comments that are as complete as possible.

On a more editorial level, we would ask you to address the following issues:

- Please provide a .docx formatted version of the manuscript text (including legends for main figures, EV figures and tables). Please make sure that the changes are highlighted to be clearly visible.
- Please provide individual production quality figure files as .eps, .tif, .jpg (one file per figure).
- Please provide a .docx formatted letter INCLUDING the reviewers' reports and your detailed point-by-point responses to their comments. As part of the EMBO Press transparent editorial process, the point-by-point response is part of the Review Process File (RPF), which will be published alongside your paper.
- Please note that all corresponding authors are required to supply an ORCID ID for their name upon submission of a revised manuscript.
- We replaced Supplementary Information with Expanded View (EV) Figures and Tables that are collapsible/expandable online (see examples in <http://msb.embopress.org/content/11/6/812>). A maximum of 5 EV Figures can be typeset. EV Figures should be cited as 'Figure EV1, Figure EV2' etc... in the text and their respective legends should be included in the main text after the legends of regular figures.

Additional Tables/Datasets should be labeled and referred to as Table EV1, Dataset EV1, etc. Legends have to be provided in a separate tab in case of .xls files. Alternatively, the legend can be supplied as a separate text file (README) and zipped together with the Table/Dataset file.

For the figures and tables that you do NOT wish to display as Expanded View figures, they should be bundled together with their legends in a single PDF file called *Appendix*, which should start with a short Table of Content. Each legend should be below the corresponding Figure/Table in the Appendix. Appendix figures and tables should be referred to in the main text as: "Appendix Figure S1, Appendix Figure S2, Appendix Table S1" etc. See detailed instructions regarding expanded view here: <https://www.embopress.org/page/journal/17444292/authorguide#expandedview>.

- Before submitting your revision, primary datasets (and computer code, where appropriate) produced in this study need to be deposited in an appropriate public database (see <https://www.embopress.org/page/journal/17444292/authorguide#dataavailability>).

The accession numbers and database should be listed in a formal "Data Availability" section (placed after Materials & Method) that follows the model below (see also <https://www.embopress.org/page/journal/17444292/authorguide#dataavailability>). Please note that the Data Availability Section is restricted to new primary data that are part of this study.

Data availability

- We would encourage you to include the source data for figure panels that show essential quantitative information. Additional information on source data and instruction on how to label the files are available at <
<https://www.embopress.org/page/journal/17444292/authorguide#sourcedata>
>.

- All Materials and Methods need to be described in the main text. We would encourage you to use 'Structured Methods', our new Materials and Methods format. According to this format, the Material and Methods section should include a Reagents and Tools Table (listing key reagents, experimental models, software and relevant equipment and including their sources and relevant identifiers) followed by a Methods and Protocols section in which we encourage the authors to describe their methods using a step-by-step protocol format with bullet points, to facilitate the adoption of the methodologies across labs. More information on how to adhere to this format as well as downloadable templates (.doc or .xls) for the Reagents and Tools Table can be found in our author guidelines: < <https://www.embopress.org/page/journal/17444292/authorguide#researcharticleguide>>. An example of a Method paper with Structured Methods can be found here: .

-Regarding data quantification:

Please ensure to specify the name of the statistical test used to generate error bars and P values, the number (n) of independent experiments (please specify technical or biological replicates) underlying each data point and the test used to calculate p-values in each figure legend. Discussion of statistical methodology can be reported in the materials and methods section, but figure legends should contain a basic description of n, P and the test applied.

Graphs must include a description of the bars and the error bars (s.d., s.e.m.).

- Please provide a "standfirst text" summarizing the study in one or two sentences (approximately 250 characters, including space), three to four "bullet points" highlighting the main findings and a "synopsis image" (550px width and 400-600px height, PNG format) to highlight the paper on our homepage.

Here are a couple of examples:

<https://www.embopress.org/doi/10.15252/msb.20199356>

<https://www.embopress.org/doi/10.15252/msb.20209475>

<https://www.embopress.org/doi/10.15252/msb.209495>

When you resubmit your manuscript, please download our CHECKLIST (<http://bit.ly/EMBOPressAuthorChecklist>) and include the completed form in your submission.

Please note that the Author Checklist will be published alongside the paper as part of the transparent process (<https://www.embopress.org/page/journal/17444292/authorguide#transparentprocess>).

If you feel you can satisfactorily deal with these points and those listed by the referees, you may wish to submit a revised version of your manuscript. Please attach a covering letter giving details of the way in which you have handled each of the points raised by the referees. A revised manuscript will be once again subject to review and you probably understand that we can give you no guarantee at this stage that the eventual outcome will be favorable.

I look forward to receiving your revised manuscript soon.

Kind regards,
Jingyi

Jingyi Hou
Editor
Molecular Systems Biology

If you do choose to resubmit, please click on the link below to submit the revision online *within 90 days*.

IMPORTANT: When you send your revision, we will require the following items:

1. the manuscript text in LaTeX, RTF or MS Word format
2. a letter with a detailed description of the changes made in response to the referees. Please specify clearly the exact places in the text (pages and paragraphs) where each change has been made in response to each specific comment given
3. three to four 'bullet points' highlighting the main findings of your study
4. a short 'blurb' text summarizing in two sentences the study (max. 250 characters)
5. a 'thumbnail image' (550px width and max 400px height, Illustrator, PowerPoint or jpeg format), which can be used as 'visual title' for the synopsis section of your paper.
6. Please include an author contributions statement after the Acknowledgements section (see <https://www.embopress.org/page/journal/17444292/authorguide>)
7. Please complete the CHECKLIST available at (<https://bit.ly/EMBOPressAuthorChecklist>). Please note that the Author Checklist will be published alongside the paper as part of the transparent process (<https://www.embopress.org/page/journal/17444292/authorguide#transparentprocess>).
8. Please note that corresponding authors are required to supply an ORCID ID for their name upon submission of a revised manuscript (EMBO Press signed a joint statement to encourage ORCID adoption). (<https://www.embopress.org/page/journal/17444292/authorguide#editorialprocess>)
9. When assembling figures, please refer to our figure preparation guideline in order to ensure proper formatting and readability in print as well as on screen:

See also figure legend guidelines: <https://www.embopress.org/page/journal/17444292/authorguide#figureformat>

Currently, our records indicate that there is no ORCID associated with your account.

Please click the link below to provide an ORCID:

Link Not Available

The system will prompt you to fill in your funding and payment information. This will allow Wiley to send you a quote for the article processing charge (APC) in case of acceptance. This quote takes into account any reduction or fee waivers that you may be eligible for. Authors do not need to pay any fees before their manuscript is accepted and transferred to the publisher.

*** PLEASE NOTE *** As part of the EMBO Press transparent editorial process initiative (see our Editorial at <https://dx.doi.org/10.1038/msb.2010.72>), Molecular Systems Biology publishes online a Review Process File with each accepted manuscripts. This file will be published in conjunction with your paper and will include the anonymous referee reports, your point-by-point response and all pertinent correspondence relating to the manuscript. If you do NOT want this File to be published, please inform the editorial office at msb@embo.org within 14 days upon receipt of the present letter.

Reviewer #1:

Review

Manuscript Reference: MSB-2021-10673

High diversity in Delta variant across countries revealed via genome-wide analysis of SARS-CoV-2 beyond the Spike protein
R. Suratekar, P. Ghosh, Michiel J.M. Niesen, G. Donadio, P. Anand, V. Soundararajan, A.J. Venkatakrishnan
ference Labs, Bengaluru 560017, Karnataka, India, nference, Cambridge, Massachusetts 02139, USA

The authors present a comprehensive and well executed analysis of mutations in delta variants of SARS-CoV-2 that arose in 176 different countries and territories. Their findings direct our attention to the possible importance of mutations in viral genome regions other than the conventionally analyzed spike protein. The latter ones have also been included in their detailed investigations. There is no clearly elaborated relation to the contribution of viral functions other than the spike protein. However, the occurrence of mutations in these non-spike segments of the viral genome highlight the need for much more profound molecular-genetic investigations of this virus. In this context, the study has identified ten prevalent mutations relating to five viral proteins among different delta variants. These proteins might be worth investigating further for their function in viral biology and pathogenicity.

How are we ever going to understand the virus's pathogenic potential without having a complete grasp of all the viral functions? Of course, the demand is easier made than fulfilled. The major merit of this report lies in challenging our very limited

understanding of SARS-CoV-2 molecular biology and genetics.

The authors have "enriched" their study by copious statistical analyses that are colorful and interesting but many of them do not necessarily contribute to a more profound understanding of viral biology. Some of the less meaningful illustrations could be omitted. Undoubtedly, it could very well be important to comprehend that there exist different mutational repertoires - as they call it - of delta variants in different regions of the globe - "country-specific core mutations".

For editorial considerations it might be diligent to place as many of the illustrations in a manuscript into the section "Supplementary Information". Nevertheless, I recommend to transfer the functionally most important ones of these Figures into the main part of the manuscript (e.g. Fig. S4, S6, S8).

The authors might be interested in a published report that has asked, in part similar, questions as they have but has also concentrated on different aspects (time course, viral proteins) of viral mutability: doi:10.15252/emmm.202114062.

In summary, the present manuscript contributes very useful information, a careful statistical evaluation of the mutational repertoire of the virus. After reading the paper, one realizes the enormous amount of missing information that we all are called upon to help decipher. I recommend to publish this manuscript after some modifications in the emphasis that the text has placed.

Reviewer #2:

Review of "High diversity in Delta variant across countries revealed via genome-wide analysis of SARS-CoV-2 beyond the Spike protein" (MSB-2021-10673) by Suratekar et al.

In this study, the authors analyzed the genome sequences of Delta variant of SARS-CoV-2 at the global scales, and examined the characteristic mutations carried by the Delta variants across countries or territories. Although the authors tabulated several key missense mutations carried by the Delta variants, it is very disappointing that the authors did not provide any exciting results other than the variants that were well characterized in GISAID or other databases such as outbreak.info or CoV-lineages. Thus, this reviewer does not think this paper reaches the standard of Molecular Systems Biology.

Reviewer #3:

Review for Manuscript: MSB-2021-10673

High diversity in Delta variant across countries revealed via genome-wide analysis of SARS-CoV-2

Very interesting and valuable work, especially in the scope of analysis.

Comments:

Results section

- page 2, first paragraph, please clarify the 6.3% value assigned to the contribution of the Spike protein gene in the unique mutations pool (as is the math is not clear)

- page 3 paragraph 1. Please define "other variants of concern".

Methods section - first paragraph, please clarify how the sequence pool contains 89,875 unique mutations, considering the complete genome has approx. 29983nt. Also how were the 8157 unique mutations considered for selection, was their presence in at least 100 sequences the only criterion? If so, then clarify the ** in Table 2 on page 14 (**unreliable conservation score due to calculations performed on less than 6 non-gapped homologous sequences?)

Page 6 - Equation one -(cosine similarity) formula is not readable

Cosine for airline connectivity - not clear if the similarity is calculated over the number of International flights or also by similarity of destinations/origins.

Figure 2 contains a lot of information. Perhaps the Venn diagram is redundant. The six mutations marked with asterisk are highly prevalent in all countries with delta variant, but nearly absent in the other variants of concern... does that mean A or all others as well? The manuscript deals exclusively with A vs D variant data.

Figure S1 is a bit hard to read and interpret. It seems that panel A shows that 99.2% of all sequences have a mutation in at least one protein, ... and 1.8% in all 24 proteins used in the analysis. Is this correct? Panel B refers to the percent of each protein's span that can acquire mutations, e.g. only 48.47% of the aa positions in the spike were found to mutate? Or is it that the spike was found to have mutations in 48.47% of the instances. It seems the latter would be incorrect, please re-word the axis

descriptors and the figure S1 text for more clarity.

Figure S2 Panel A is not explained well. It is not clear what are the 12 mutations high, highlighted in magenta, please clarify if the X axis is the same for both panels.

Figure S4 The geographical distribution of the 4 clusters is well illustrated and noted, but not explained in the discussion. In fact, FS5 does not help to explain this phenomenon, since it appears to claim that air connectivity has not affected the Delta variant diversity.

Figure S5 Panel C is not explained. What is the difference between the two graphs?

Figure S8 panel B appears to diagram the overlap between the 4 different variants, not the country specific mutations per variant?

Table S1 What was the cut off for high prevalence

Based on these observations minor revision is recommended

Response to reviewers

We thank the reviewers and greatly appreciate their valuable comments, suggestions, and feedback on our manuscript. Addressing these comments has helped strengthen the manuscript significantly.

Reviewer #1

The authors present a comprehensive and well executed analysis of mutations in delta variants of SARS-CoV-2 that arose in 176 different countries and territories. Their findings direct our attention to the possible importance of mutations in viral genome regions other than the conventionally analyzed spike protein. The latter ones have also been included in their detailed investigations. There is no clearly elaborated relation to the contribution of viral functions other than the spike protein. However, the occurrence of mutations in these non-spike segments of the viral genome highlight the need for much more profound molecular-genetic investigations of this virus. In this context, the study has identified ten prevalent mutations relating to five viral proteins among different delta variants. These proteins might be worth investigating further for their function in viral biology and pathogenicity.

How are we ever going to understand the virus's pathogenic potential without having a complete grasp of all the viral functions? Of course, the demand is easier made than fulfilled. The major merit of this report lies in challenging our very limited understanding of SARS-CoV-2 molecular biology and genetics.

The authors have "enriched" their study by copious statistical analyses that are colorful and interesting but many of them do not necessarily contribute to a more profound understanding of viral biology. Some of the less meaningful illustrations could be omitted. Undoubtedly, it could very well be important to comprehend that there exist different mutational repertoires - as they call it - of delta variants in different regions of the globe - "country-specific core mutations".

For editorial considerations it might be diligent to place as many of the illustrations in a manuscript into the section "Supplementary Information". Nevertheless, I recommend to transfer the functionally most important ones of these Figures into the main part of the manuscript (e.g. Fig. S4, S6, S8).

Response: We thank the reviewer for their feedback. As suggested, the previous Figures S3, S7, S9, and Tables S1-S3 have been moved out of the Supplementary section to the Appendix. We have also transferred the previous Figure S4 into the main manuscript as Figure 3. However, since Figures S6 (**Figure EV4**) and S8 (**Figure EV5**) focus more on technical details, we have retained those as Expanded View figures.

The authors might be interested in a published report that has asked, in part similar, questions as they have but has also concentrated on different aspects (time course, viral proteins) of viral mutability: doi:10.15252/emmm.202114062.

Response: We thank the reviewer for pointing to this paper. We now cite this study in the Discussion section of the revised version of our manuscript:

“These differences contribute to the risk of emergence of new SARS-CoV-2 variants, which could pose challenges to existing therapies and vaccination⁴⁰.” (Page 5)

In summary, the present manuscript contributes very useful information, a careful statistical evaluation of the mutational repertoire of the virus. After reading the paper, one realizes the enormous amount of missing information that we all are called upon to help decipher. I recommend to publish this manuscript after some modifications in the emphasis that the text has placed.

Response: We again thank the reviewer for their feedback.

Reviewer #2

In this study, the authors analyzed the genome sequences of Delta variant of SARS-CoV-2 at the global scales, and examined the characteristic mutations carried by the Delta variants across countries or territories. Although the authors tabulated several key missense mutations carried by the Delta variants, it is very disappointing that the authors did not provide any exciting results other than the variants that were well characterized in GISAID or other databases such as outbreak.info or CoV-lineages. Thus, this reviewer does not think this paper reaches the standard of Molecular Systems Biology.

Response: We thank the reviewer for their feedback. Indeed, there are databases such as GISAID and outbreak.info that characterize SARS-CoV-2 variants. We would like to clarify though that the differentiator of our study is that it has highlighted the country-specific differences in the Delta variant by accounting for mutations across all the 26 SARS-CoV-2 proteins, which we believe is important and underscores the gaps in our understanding, as highlighted by Reviewer 1 and Reviewer 3.

Reviewer #3

Very interesting and valuable work, especially in the scope of analysis.

Comments:

Results section

- page 2, first paragraph, please clarify the 6.3% value assigned to the contribution of the Spike protein gene in the unique mutations pool (as is the math is not clear)

Response: We thank the reviewer for this comment. The Spike protein harbors mutations in 617 amino acid positions out of the total 9757 amino acids in the SARS-CoV-2 proteome, which accounts for the 6.3% value. We have now clarified the math in the revised manuscript:

“The 1055 unique amino acid mutations across 617 positions in the Spike protein contributes to only 6.3% of the mutated SARS-CoV-2 proteome (617 mutated positions out of the total 9757 amino acids in the SARS-CoV-2 proteome). This emphasizes the need to study the mutational profile across all the proteins of SARS-CoV-2.” (Page 2)

- page 3 paragraph 1. Please define "other variants of concern".

Response: We thank the reviewer for this comment. 'Other variants of concern' refer to the Alpha, Beta, and Gamma variants of SARS-CoV-2. We have now clarified this point in the revised manuscript:

“Strikingly, all these mutations except P323L in NSP12 are nearly exclusive to the Delta variant compared to other variants of concern (Alpha, Beta, and Gamma variants of SARS-CoV-2)...” (Page 3)

Methods section - first paragraph, please clarify how the sequence pool contains 89,875 unique mutations, considering the complete genome has approx. 29983nt.

Response: We thank the reviewer for raising this concern. The 89,875 mutations are amino acid mutations. We have now clarified this in the revised manuscript:

“1,986,688 sequences harbor a total of 89,875 unique amino acid mutations.” (Page 6)

The 29,983 nucleotides code for 9757 amino acids in total, which can each be theoretically substituted to 19 other amino acids, deleted or new amino acids can be inserted. Of these, 89,875 amino acid mutations have been reported in the GISAID database until 31 July 2021.

Also how were the 8157 unique mutations considered for selection, was their presence in at least 100 sequences the only criterion? If so, then clarify the ** in Table 2 on page 14 (**unreliable conservation score due to calculations performed on less than 6 non-gapped homologous sequences?)

Response: We thank the reviewer for this comment. As rightly pointed out by the reviewer, the 8157 unique mutations are those present in 100 or more SARS-CoV-2 sequences in the GISAID database.

However, in Table 2 we have analyzed SARS-CoV-2 homologous sequences from the UniRef90 database using ConSurf (<https://consurf.tau.ac.il/>)¹, which are sets of sequences from UniProtKB and selected UniParc records, clustered based on at least 90% sequence identity to, and an 80% overlap with, the seed sequence (the longest sequence in the cluster).

1. Ashkenazy H, Abadi S, Martz E, et al. ConSurf 2016: an improved methodology to estimate and visualize evolutionary conservation in macromolecules. *Nucleic Acids Res.* 2016;44(W1):W344-W350.

Thus in contrast to the analyzed GISAID database with only SARS-CoV-2 sequences, the searches against the UniRef90 have identified homologous sequences from other coronaviruses as well (**Table 2**). The ** positions might arise due to it being limited to SARS-CoV-2 and rarer in its homologs.

We have now clarified this in the revised version of the manuscript:

“We have assessed the evolutionary conservation of the ten characteristic Delta variant mutations using ConSurf — graded on a scale of 1 (variable) to 9 (conserved) (Table 2). Protein sequence homologs were retrieved using HMMER against the UniRef90 database, and the multiple sequence alignment was built using MAFFT.” (Page 3)

“We retrieved 1,987,504 SARS-CoV-2 high coverage complete genome sequences from human hosts in 176 countries/territories spanning 1336 PANGO lineages on 18 August 2021 from GISAID for December 2019 - July 2021, of which 816 sequences do not harbor any mutations. We removed sequences from other hosts and those with incomplete dates (YYYY-MM or YYYY) from further analyses. 1,986,688 sequences harbor a total of 89,875 unique amino acid mutations. However, to account for errors arising from sequencing, we only consider 8157 unique mutations in 24 proteins that are present in 100 or more sequences for all our further analyses.” (Page 6)

Page 6 - Equation one -(cosine similarity) formula is not readable

Response: We thank the reviewer for this suggestion. We have now increased the font size for both equations 1 and 2. (**Page 6**)

Cosine for airline connectivity - not clear if the similarity is calculated over the number of International flights or also by similarity of destinations/origins.

Response: We thank the reviewer for this comment. The airline connectivity is calculated only over the number of international flights between two countries (origin/destination). We have now clarified this in the revised manuscript:

“A matrix of the number of international flights across all countries of the world was generated for the period of February 2021 - June 2021.” (Page 7)

Figure 2 contains a lot of information. Perhaps the Venn diagram is redundant. The six mutations marked with asterisk are highly prevalent in all countries with delta variant, but nearly absent in the other variants of concern... does that mean A or all others as well? The manuscript deals exclusively with A vs D variant data.

Response: We thank the reviewer for this comment. We have now removed the Venn diagram from Figure 2. Also, the mutations marked in asterisk are highly prevalent in all countries with the Delta variant, but nearly absent from the Alpha, Beta, and Gamma variants of SARS-CoV-2. We have now clarified this in the revised manuscript:

“The six mutations (in other SARS-CoV-2 proteins) marked with an asterisk are highly prevalent in all countries of occurrence of Delta variant (mean prevalence = 99.74%) but are nearly absent (mean prevalence = 0.12%) in the other variants of concern (Alpha, Beta, and Gamma variants of SARS-CoV-2).” (Figure 2 legend)

Figure S1 is a bit hard to read and interpret. It seems that panel A shows that 99.2% of all sequences have a mutation in at least one protein, ... and 1.8% in all 24 proteins used in the analysis. Is this correct?

Response: We thank the reviewer for raising this concern. Panels A and B in Figure S1 had a shared x-axis. It means that 99.2% of all sequences analyzed have at least one mutation in the Spike protein and 1.8% of all sequences analyzed have at least one mutation in the NS6 protein. However, for clarity, we have now labeled the x-axis in panel A as well (**Figure EV1**).

Panel B refers to the percent of each protein's span that can acquire mutations, e.g. only 48.47% of the aa positions in the spike were found to mutate? Or is it that the spike was found to have mutations in 48.47% of the instances. It seems the latter would be incorrect, please re-word the axis descriptors and the figure S1 text for more clarity.

Response: We thank the reviewer for raising this concern. As rightly pointed out by the reviewer, it means that only 48.47% of the amino acid positions in the Spike protein were found to be mutated. We have now clarified this in the revised manuscript:

“(B) Percentage of SARS-CoV-2 protein sequence lengths that can acquire mutations. The NS7a and NS8 are hypervariable proteins that mutate 92.56% and 90.08% of their sequence lengths.” (Figure EV1 legend)

Figure S2 Panel A is not explained well. It is not clear what are the 12 mutations high, highlighted in magenta, please clarify if the X axis is the same for both panels.

Response: The 12 mutations (highlighted in magenta) are highly prevalent in all 104 countries of occurrence of the Delta variant. As rightly pointed out by the reviewer, the x-axis is shared across the two panels. However, for clarity, we have now labeled the x-axis in panel A as well (**Figure EV2**).

Figure S4 The geographical distribution of the 4 clusters is well illustrated and noted, but not explained in the discussion. In fact, FS5 does not help to explain this phenomenon, since it appears to claim that air connectivity has not affected the Delta variant diversity.

Response: We thank the reviewer for this suggestion. We have now explained it in the discussions section of the revised manuscript:

“Our study also motivates that the diversity at the proteome level should be considered in designating the variants of concern and interest. This study shows that the sub-variants of the Delta variant are prevalent in geographically distant countries (Fig 3), eliminating a causal relationship of geographical proximity with Delta variant diversity. However, future studies are warranted to comprehensively examine the combinations of factors such as vaccination rates, geographical proximity, and airline connectivity (Fig EV3) to dissect the difference in the epidemiology of Delta variants across countries.” (Page 5)

However, as shown in Figure S5 (**Figure EV3**) airline connectivity does not have a causal relationship to the diversity of the Delta variant. The current Figure EV3 is updated to include airline connectivity data from all 104 countries in which the Delta variant was detected.

Figure S5 Panel C is not explained. What is the difference between the two graphs?

Response: We thank the reviewer for raising this concern, and apologize for the error. We have now clarified this in the revised manuscript:

“(C) Frequency histogram for the distribution of the cosine similarity values between country-specific core mutations in Delta variant across all countries of its occurrence (left) and airline connectivity across countries (right).” (Figure EV3 legend)

Figure S8 panel B appears to diagram the overlap between the 4 different variants, not the country specific mutations per variant?

Response: The union set of country-specific core mutations was calculated for the Alpha, Beta, Gamma, and Delta variants of SARS-CoV-2, and their overlap is shown in **Figure EV5 panel B**: *“A union set of country-specific core mutations from all countries in which lineage L is present were also determined. We observed that the Delta variant’s union set of country-specific core mutations are distinct and higher from those in the other variants of concern (Fig EV5, Appendix Table S3)” (Page 8)*

Table S1 What was the cut off for high prevalence

Response: By “highly prevalent”, we are indicating mutations that are present in all 104 countries of occurrence of the Delta variant and have a mean prevalence of 99% or more. We have now clarified this in the revised manuscript:

“(B) Mean percent prevalence of Delta mutations. The 12 mutations highlighted in magenta are highly prevalent (mean prevalence > 99%) in all countries of occurrence of the Delta variant (Appendix Table S1).” (Figure EV2 legend)

Based on these observations minor revision is recommended

Response: We again thank the reviewer for their feedback.

16th Dec 2021

Manuscript Number: MSB-2021-10673R

Title: High diversity in Delta variant across countries revealed via genome-wide analysis of SARS-CoV-2

Author: A J Venkatakrisnan

Rohit Suratekar

Pritha Ghosh

Michiel Nielsen

Gregory Donadio

Praveen Anand

Venky Soundararajan

Thank you again for submitting your work to Molecular Systems Biology. We are overall satisfied with the revisions made. Before we can formally accept your manuscript, we would ask you to address the following editorial-level issues.

Kind regards,
Jingyi

Jingyi Hou
Editor
Molecular Systems Biology

If you do choose to resubmit, please click on the link below to submit the revision online before 15th Jan 2022.

IMPORTANT: When you send your revision, we will require the following items:

The authors have made all requested editorial changes.

7th Jan 2022

Manuscript number: MSB-2021-10673RR

Title: High diversity in Delta variant across countries revealed by genome-wide analysis of SARS-CoV-2

Dear Dr Venkatakrishnan,

Thank you again for sending us your revised manuscript. We are now satisfied with the modifications made and I am pleased to inform you that your paper has been accepted for publication.

*** PLEASE NOTE *** As part of the EMBO Publications transparent editorial process initiative (see our Editorial at <https://dx.doi.org/10.1038/msb.2010.72>), Molecular Systems Biology publishes online a Review Process File with each accepted manuscripts. This file will be published in conjunction with your paper and will include the anonymous referee reports, your point- by-point response and all pertinent correspondence relating to the manuscript. If you do NOT want this File to be published, please inform the editorial office at msb@embo.org within 14 days upon receipt of the present letter.

Should you be planning a Press Release on your article, please get in contact with msb@wiley.com as early as possible, in order to coordinate publication and release dates.

LICENSE AND PAYMENT:

All articles published in Molecular Systems Biology are fully open access: immediately and freely available to read, download and share.

Molecular Systems Biology charges an article processing charge (APC) to cover the publication costs. You, as the corresponding author for this manuscript, should have already received a quote with the article processing fee separately. Please let us know in case this quote has not been received.

Once your article is at Wiley for editorial production you will receive an email from Wiley's Author Services system, which will ask you to log in and will present you with the publication license form for completion. Within the same system the publication fee can be paid by credit card, an invoice or pro forma can be requested.

Payment of the publication charge and the signed Open Access Agreement form must be received before the article can be published online.

Molecular Systems Biology articles are published under the Creative Commons licence CC BY, which facilitates the sharing of scientific information by reducing legal barriers, while mandating attribution of the source in accordance to standard scholarly practice.

Proofs will be forwarded to you within the next 2-3 weeks.

Thank you very much for submitting your work to Molecular Systems Biology.

Sincerely,
Jingyi

Jingyi Hou
Editor
Molecular Systems Biology

Corresponding Author Name: A. J. Venkatakrishnan, Venky Soundararajan

Manuscript Number: MSB-2021-10673